# Integrating a Health Literacy Lens into Nutrition Labelling Policy in Canada

**DOI:** 10.3390/ijerph17114130

**Published:** 2020-06-10

**Authors:** Elizabeth Mansfield, Rana Wahba, Elaine De Grandpré

**Affiliations:** Bureau of Nutritional Sciences, Food Directorate, Health Canada, Ottawa, ON K1A 0K9, Canada; rana.wahba@canada.ca (R.W.); elaine.degrandpre@canada.ca (E.D.G.)

**Keywords:** health literacy, nutrition labelling, policy, regulations

## Abstract

An increasingly important concern in public health practice is health literacy. Simply stated, it refers to the interactions between individuals and health related information to make informed decisions concerning their health. Research shows that consumers face many health literacy challenges in accessing, understanding and evaluating nutrition labelling information when making food choices. The systematic integration of health literacy considerations into social science and consumer behaviour research can help address these challenges and better meet the needs of the increasingly diverse Canadian population. This application of a health literacy lens should be considered for all future food and nutrition labelling research, to maximize the positive impact of subsequent health policies and regulations on health outcomes and health status of Canadians.

## 1. Introduction

Canada’s ethnocultural diversity and the challenges for literacy and health literacy as they relate to informed food choices is an underestimated problem. In addition, an unhealthy diet is one of the top risk factors for obesity and chronic disease burden in Canada [1]. Consumer use of nutrition information on food labels to make informed food choices, to plan healthy meals and to manage diseases of public health concern impacted by diet requires a range of health literacy skills [2]. This unique set of cognitive and social skills reflects the knowledge, motivation and competencies of individuals to make use of appropriate nutrition information in ways that promote and maintain health [3]. Not surprisingly, health literacy is increasingly being seen as an important contributor to the health of Canadians, and is linked to health disparities [4].

Nutbeam’s tripartite model of health literacy links health literacy to people’s “knowledge, motivation and competencies to access, understand, appraise, and apply health information in order to make decisions in everyday life concerning healthcare, disease prevention and health promotion, to maintain or improve quality of life during the life course” [5] (Table 1). This tripartite model highlights the importance of achieving health literacy at functional, interactive and critical levels. At the most basic level, functional nutrition literacy involves the basic set of reading and writing skills to be able to function in the everyday food shopping environment, such as engaging with nutrition labelling to be able to identify foods high in saturated fat, sodium and/or sugars. Interactive health literacy requires a more developed set of cognitive and literacy skills to interpret, evaluate and use the requisite nutrition label information to self-manage specific dietary goals, or to reduce disease risk with the selection of healthier foods. Critical health literacy encompasses critical nutrition label appraisal skills and the use of food skills (e.g., the use of seasonal or frozen vegetables) to address barriers to healthy eating at the community level [6]. Health literacy is also a systemic concern, and is the responsibility of systems through which the relevant health information is provided.

An effective nutrition labelling policy requires a focus not only on the nutrition information being conveyed, but also on the varied contexts in which consumers engage with it. In addition, for nutrition labelling to be effective, the requisite information must be noticed, processed, evaluated and used by those at the greatest risk of limited/marginal health literacy [7]. Policy efforts to increase the effectiveness of food and health information systems in supporting healthy eating practices and decreasing disparities in population health may be limited if these health literacy skills are not addressed [8]. By improving how the food system provides nutrition label information, and improving people’s access and capacity to effectively use that information, both the individuals and systems become more health literate at functional, interactive and critical levels [9]. Diet equity, as it relates to supporting informed food choices and healthier dietary intakes, may also be advanced [7].

In this paper, we introduce the integration of a health literacy lens as an analytical process used to assess how those groups of Canadians at risk of limited or marginal health literacy may be impacted by proposed federal health policy initiatives. We discuss the systematic integration of health literacy considerations into social science and consumer behaviour research, to better address the health literacy needs of the increasingly diverse Canadian population, and improve the efficiency of the investigative process. The overall goal is to increase the capacity of researchers to adopt scientific practices integrating a health literacy lens, where appropriate, into health-related research, to maximize the positive impact of subsequent health policies on health outcomes and the health status of their populations of interest.

## 2. Health Literacy Research Considerations

While nutrition labelling on food packages is the most reported source of nutrition information guiding consumers’ food selections [10], shopping for foods has become an increasingly complex task. Consumers are exposed to a vast array of foods with a large range of price and product choices, and also interact with a variety of regulated labelling attributes and food industry marketing messages on food packages when making their food decisions. It is not surprising that food label use varies considerably [11,12,13], likely due, in part, to the challenges consumers face in accessing, understanding, evaluating and using nutrition label information appropriately to make their decisions [14].

Recent Canadian research highlights the need for a research strategy that supports the development and refinement of nutrition labelling policies and regulations to meet the needs of all Canadians, and more importantly, those facing constraints due to limited/marginal health literacy [11]. The measurement of health literacy levels as a moderating variable for consumer use of nutrition information when making food choices is noticeably absent in nutrition labelling research. The previous use of socio-economic status and level of education as proxies for health literacy are not appropriate for enhancing our understanding of health literacy in nutrition outcomes [15]. In this section, we present different health literacy considerations and demonstrate the value of a deliberate and systematic integration of a health literacy lens into social science research, supporting the development of nutrition labelling policy and regulations.

### 2.1. Adaptation of a Health Literacy Assessment Tool

Situating health literacy as a population asset in today’s technology-rich food and nutrition environment requires effective measurement tools. Consumers must be able to use and understand both text and numbers, if they are to successfully engage with nutrition information on food labels in making their food choices. The Newest Vital Sign (NVS), a validated health literacy assessment tool, assesses a person’s likelihood of limited, marginal or adequate health literacy [16]. Among the different health literacy assessment tools available, the NVS has specific utility for food and nutrition research, because it is framed in a food context. The tool assesses prose (text) and numeracy skills using a nutrition facts table and a list of ingredients on a container of ice cream. It is also quick and easy to administer (6 questions in about 3 minutes). In a recent systematic review of the role of health literacy in predicting adherence to nutritional recommendations, studies that used the NVS reported a significant direct positive association of health literacy and dietary quality, and a significant direct negative association with sugar-sweetened beverages intake within the general population [17]. The focus of the NVS on functional aspects of reading and numeracy in the context of a food label addresses the content and context specific domains of health literacy [18] and highlights its value for use in nutrition labelling research. Furthermore, the use of a nutrition label to assess health literacy is intuitively appealing, because nutrition labels are familiar across socio-demographics, and are an important part of health management for many chronic diseases [19,20].

Health Canada adapted the NVS for use in Canada as a web-based, bilingual (English, French) computerized tool [21]. The electronic interface integrates audio to deliver all written information (i.e., each of the 6 questions and each of the multiple-choice options for each question), a format that does not require subjects to read the instructions or questions [21]. The tool also includes an introductory slide that walks the participant through all interactive elements on the screen. It explains which onscreen icon to click to listen to the onscreen information, where to locate and choose an answer, how to change an answer and where to click to move on to the next question. This is done in an active way, meaning the participant is asked to carry out each set of onscreen instructions, as the voice recording states each of them. The benefit of an electronic interface is the minimization of the acknowledged discomfort or stigmatization that individuals sometimes report with interviewer-based tools, especially among those at highest risk of limited health literacy [22]. The computerized NVS can be integrated into consumer research as a health literacy screening tool for participant assessment and recruitment purposes [23,24,25].

### 2.2. Applying a Health Literacy Lens into Participant Recruitment

The Canadian adaptation of the NVS [21] supports a recruitment strategy for qualitative and quantitative consumer-based research, which generates a primary pool of research participants reflecting the health literacy level of Canadians (approximately 60% marginal vs. 40% adequate) [26]. This extends the consumer research capacity beyond the common considerations of low education and low income, to address the health literacy needs of the increasingly diverse Canadian population, with interventions aimed at improving their knowledge and understanding [11].

Of recent interest are the established national online panel providers that have been used by Canadian researchers to conduct consumer surveys and test the efficacy of nutrition labelling approaches, including warning labels on sugar-sweetened beverages [27], the effectiveness of front of pack labelling [28], perceptions of the nutrition facts table and front of pack labelling [29], as well as comprehension and use of nutrition information [30], and specifically calorie information on pre-packaged foods [31]. These national panels offer great potential for quantitative research data collection as they may easily contain over 400,000 Canadians diverse in age, gender, language and region. The generalizability of data collected with online panels, however, is uncertain. The integration of the Canadian adaptation of the NVS [21] as a screening tool within these market research panels has the capacity to generate a representative sample of the Canadian population in terms of health literacy status, gender, age, first language and province/territory of residence [23].

### 2.3. Integrating a Health Literacy Lens into Research Methods and Analysis

The integration of a health literacy lens into the development of consumer research methods extends the user-centered research paradigm by tasking consumers with objective tests of awareness, understanding and appraisal of labelling components, rather than relying on less robust self-reported use of labelling attributes. Interviewer guides, moderator guides and survey questions developed using plain language principles ensures that all participants, especially those at risk of marginal health literacy, are more likely to understand what is being asked of them [32]. This means using simple and common words, the active voice and simple and short sentences. It includes the use of direct statements, omitting the inclusion of irrelevant background information and adaptation of a communication strategy, based on each participant’s health literacy level. Using a slower speech rate, pausing at the end of each sentence and informing the participants that all information can be repeated as needed, are a few important elements to adhere to with participants at risk of marginal health literacy. Giving examples is beneficial when explaining new or complex topics. This practice also applies when developing graphic images and/or mock food packages to be used in experiential research with consumers. For example, the format of the packaged food label, including the way text and any graphics are displayed on it, should not impede consumer understanding. Pilot testing of research tools with consumers of varying health literacy levels helps determine the best format for a question (i.e., Likert scale vs. ranking vs. multiple-choice), as well as a key set of multiple-choice derivatives for select questions. For qualitative research, pilot testing with consumers of varying health literacy levels helps to refine the presentation, wording and format of key questions, as well as to identify pertinent probing questions more likely to be understood by those with limited/marginal health literacy.

It is well known that stigma associated with low health literacy can impair one’s spoken interactions, particularly when encountering health related issues and information [33]. The organization of discussion groups that considers the health literacy level of participants permits the appropriate grouping of participants. This can help minimize the potential stigma and discrimination some people might experience in a group discussion setting. Grouping by health literacy level can help remove the often-hidden barrier and discomfort associated with low literacy. It also encourages group moderators to be more intentional in effective communication using plain language. The complexity of explanations and instructions participants encounter throughout the discussion group process can be minimized, allowing for more open communication. Moreover, the integration of health literacy as a screening tool for larger scale research trials eliminates the risk of low awareness by research recruitment staff of the potential for low literacy among prospective participants. For online surveys and web-based trials, computer skills and reading skills can be significant barriers to participation. To address these barriers, audio can be integrated into the entirety of the online consumer research tool, including voice-overs for each question (and multiple-choice answers where applicable), which can be listened to as needed. This integration of a health literacy lens better addresses the health literacy needs of all research participants, and improves the efficiency of the investigative process.

A recent systematic review highlights the need to explore the moderating and mediating roles of an individual’s health literacy status on nutrition outcomes [15]. In qualitative research, thematic content analysis by health literacy level (marginal vs. adequate) using core competencies of accessing, understanding, and appraising labelling information provides more nuanced information on the salience of labelling attributes and approaches. Quantitative research analysis with health literacy as a moderating variable identifies gaps in awareness, understanding and appraisal of the efficacy of specific labelling attributes and approaches. Including health literacy as a moderating variable also helps to better understand how health literacy impacts consumer competencies in the context of food decisions.

Overall, this constructive and systematic application of a health literacy lens into social science and consumer behaviour research fills in critical gaps and opportunities to inform the development of nutrition policies that address the health literacy needs of the increasingly diverse Canadian population. The integration of health literacy as an analytical process can improve research practices through the development of modern research tools and methods that are optimally conducive to robust, reproducible and generalizable nutrition related health outcomes.

## 3. Conclusions

Food labels are a key source of credible nutrition information for Canadians to make informed food choices, plan nutritious meals and manage chronic diseases and conditions impacted by diet [11]. There is a paucity of research examining the role health literacy plays in consumers’ use of nutrition labelling to make food choices. Indeed, for nutrition labelling to be effective, the requisite information must be noticed, processed, evaluated and used by those at greatest risk of limited/marginal health literacy [7]. Policy efforts to increase the effectiveness of food and health information systems in supporting healthy eating practices, and decreasing disparities in population health may be limited if these health literacy skills are not addressed [8]. By including health literacy, a social determinant of health, in the development of these nutrition labelling policies, diet equity, as it relates to supporting informed food choices and healthier dietary intakes, may also be advanced [7].

Canada’s ethnocultural diversity and the challenges for literacy and health literacy as they relate to informed food choice is an underestimated problem. When practitioners, policymakers and health researchers adopt practices that assume significant literacy skills from their participants, the effectiveness for a large and often well-hidden population of people at risk of marginal/limited health literacy is limited [33]. Efforts to address this diversity and improve health literacy will help people use health information to improve their quality of life and effectively create a more inclusionary society in Canada.

Policymakers, practitioners and researchers need to collaborate on interventions that integrate a health literate approach to empower consumers to access, understand, appraise and apply nutrition information to make informed food choices in a range of multicultural settings, including in-store, online and outside the home in restaurants and other food services. The testing and evaluation of nutrition labelling initiatives to improve food choices and eventual health outcomes should focus on the acceptability of interventions to people at risk of limited/marginal health literacy levels, and not only those likely to have adequate literacy levels. The integration of a health literacy lens into key components of nutrition labelling consumer research within a multicultural society, including participant recruitment, research tool development, study design and data analysis, ensures that nutrition labelling policy and regulatory levers better meet the needs of Canadians of varying health literacy levels. The goal is to maximize the positive impact on health outcomes and health status of Canadians.

## Figures and Tables

**Table 1 ijerph-17-04130-t001:** The Health Literacy Process in Food Based Decision-Making.

Access Ability to find, seek and obtain appropriate food and nutrition information when making food choices
Understand Ability to comprehend the accessed food and nutrition information in the context of healthier food choices
Appraise Ability to interpret, filter, judge and evaluate the accessed food and nutrition information in the context of personal/family dietary goals and needs
Apply Ability to communicate and use the accessed food and nutrition information to make a food choice to meet dietary goals and needs

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
