# Peer review of "Integrating a Health Literacy Lens into Nutrition Labelling Policy in Canada"

_ijerph, 2020, doi:10.3390/ijerph17114130_

Round 1

Reviewer 1 Report

The manuscript by Mansfield et al describes the application of a health literacy lens in nutrition research for policy. Health literacy undoubtedly plays an important role in the various types of studies presented and the authors provide a number of common sense arguments in favor of health literacy assessment.

What appears to be missing is an objective demonstration of the added value of using a health literacy lens, be it more robust data, more effective nutrition/health policy, or other such clear benefits. Ideally this would be complemented with examples of ineffective policy or other clear cases of failure due to the lack of a health literacy lens.

Unless the above can be provided, the article should probably be identified as a commentary or opinion piece.

As a minor comment, the rarely used word parsity (l. 68) seems out of place in an article about health literacy. Paucity, scarcity, or simply saying that "Research ... is scarce" or "There is little research" are better options in my view.

Last, references 1-3 need revising so that IHME and Health Canada are not abbreviated but spelled out in full.

Author Response

Response to Reviewer 1 Comments

Point 1: What appears to be missing is an objective demonstration of the added value of using a health literacy lens, be it more robust data, more effective nutrition/health policy, or other such clear benefits. Ideally this would be complemented with examples of ineffective policy or other clear cases of failure due to the lack of a health literacy lens.

Response 1: We revised the paper to better focus on the application of a health literacy lens as an analytical process used to asses how groups of Canadians at risks of marginal or limited health literacy may be impacted by federal health policy initiatives. We removed Section 2.1 and merged Section 2.2 in with Section 3 under a new Section 2 titled "health Literacy Considerations" to better address the added value of using a health literacy lens.

Point 2: As a minor comment, the rarely used word parsity (l. 68) seems out of place in an article about health literacy. Paucity, scarcity, or simply saying that "Research ... is scarce" or "There is little research" are better options in my view.
Response 2: We replaced parsity with the word paucity

Point 3: Last, references 1-3 need revising so that IHME and Health Canada are not abbreviated but spelled out in full.

Response 3: We updated the references so that the terms are not abbreviated

Reviewer 2 Report

Thank you for your work on this interesting manuscript. It is wonderful to see efforts being put toward the application of health literacy research in health policy. Below are my thoughts:

- Most articles I have reviewed throughout my career have been original research studies focused on a narrow, but important, topic. This article is much different because it looks broadly at the idea of health literacy and nutrition, and simultaneously 1) recommends methodological steps to be taken by researchers, 2) reports on how data has been collected in Canada, and 3) suggests next steps for the field of health literacy and nutrition in the country. This multi-pronged approach left me a bit confused throughout. I was not sure if I was reading a theoretical paper, an original research article, or a policy report. Therefore, my primary feedback is for the authors to clearly articulate the purpose of your manuscript to the readers. Then, with a clear focus, eliminate unnecessary information and stick to the task at hand. If the primary focus is to report on what has been done in Canada and make recommendations for the future, stick exclusively to that aim. Likewise, if your intent is to provide a report on the development and execution of a large-scale study, stick exclusively to that aim. Simply put, I was lost as to the direction of the manuscript, and thus I think its effectiveness in conveying its intended message was minimized. 

Author Response

Point 1: My primary feedback is for the authors to clearly articulate the purpose of your manuscript to the readers. Then, with a clear focus, eliminate unnecessary information and stick to the task at hand.

Response 1: We revised the paper to focus on the integration of a health literacy lens as an analytical process used to assess how those groups of Canadians at risks of limited or marginal health literacy may be impacted by proposed federal health policy initiatives.

We included the overall goal or the paper: "is to increase the capacity of researchers to adopt scientific practices integrating a health literacy lens, where appropriate, into health related research to maximize the positive impact of subsequent health policies on health outcomes and health status of their populations of interest."

We revised sections 2 and 3 into one section that discusses the systematic integration of health literacy considerations into social science and consumer behaviour research to better address the health literacy needs of the increasingly diverse Canadian population and improve the efficiency of the investigative process. For each initiative that we introduce in this section we discuss how it is proving to be beneficial in improving nutrition research and analysis capacity while strengthening the understanding of the benefits of this deliberate and systematic application of a health literacy lens into nutrition policy and regulatory design.

We removed section 4 which discussed examples of Health Canada’s  integration of health literacy considerations into social science and consumer behaviour research supporting the development of nutrition labelling policy and regulations. With the tighter focus of the paper this section is no longer required.

Round 2

Reviewer 1 Report

I thank the authors for addressing my comments so comprehensively. Some errors (typos, grammar) have been introduced inadvertently in the revision and would need fixing before publication.

Author Response

The manuscript has been edited for typographical errors and grammar as requested.

Reviewer 2 Report

The changes made by the authors were better than expected. The purpose of the paper is clear, and the reframed and restructured presentation makes it much more reader friendly. 

Author Response

Spell check completed on revised manuscript as requested